# OpenReview forum: "Detecting and Filtering Unsafe Training Data via Data Attribution with Denoised Representation"
_ICML.cc/2026/Conference — ICML 2026 regular_

### Official Review · Reviewer_dh4A · 2026-03-09

**Soundness:** 2
**Presentation:** 3
**Significance:** 2
**Originality:** 2
**Overall Recommendation:** 4
**Confidence:** 3

**Summary:**

This paper studies unsafe training-data detection for LLMs and proposes to use the representation of the training data to measure the similarity with a target set. The method centers and whitens training/target representations using background statistics, then selects a subset of PCA directions by maximizing a leave-target-out discriminability objective. It is applied on top of GradSim and RepSim for jailbreak-data detection and a gender-bias mitigation setting.

**Compliance With Llm Reviewing Policy:**

Affirmed.

**Key Questions For Authors:**

1. How is the performance of selecting unsafe training data using powerful LLMs like GPT-5 or Gemini Pro? And compare the performance with your method.
2. Maybe provide some examples of the selection of unsafe data
3. For Dimension reduction, do you try any other non-linear dimension reduction methods other than PCA?

**Limitations:**

See above.

**Strengths And Weaknesses:**

Pros:
1. The presentation is pretty clear and easy to follow.
2. The method is simple enough to be usable, and the empirical improvements are substantial across both explicit jailbreak data and a subtler bias setting.

Cons:
1. The core detector is a centered, whitened correlation / Mahalanobis-style score motivated by classical linear signal detection, followed by subset selection. That is a reasonable idea, but the experiments do not compare against nearby simpler controls such as centered cosine similarity, whitening-only scoring, or other covariance-aware linear baselines.
2. The downstream evaluation is incomplete. In the jailbreak experiments, lower ASR could partly reflect broader refusal behavior or degraded benign utility. In the bias experiments, reporting only TPR gender gap is not enough, because the gap can shrink even if both groups’ TPRs get worse.
3. The theoretical contribution is quite weak.

---

> ### Author Rebuttal · Authors · 2026-03-30
>
> > Comparison against nearby simpler methods
>
> We thank the reviewer for this comment. These nearby controls are in fact already covered in our experiments. The raw GradSim / RepSim baselines correspond to the simple centered similarity-style scoring on the original representation space. In addition, Appendix E includes a whitening-only control (“Only Denoising”  in Table 7), which applies whitened PCA without the subsequent dimension-selection stage. This control yields clear gains over the raw baseline, but remains consistently below full DRA across models and datasets. This evidence shows that the empirical gain of DRA is attributable to the full pipeline rather than to any single step.
>
> > The downstream evaluation is incomplete, utility might degrade as ASR drops.
>
> | Method       | XSTest | ToxicChat | Gender-bias |
> |--------------|-------:|----------:|------------:|
> | No Removal   | 0.642  | 0.646     | 0.651       |
> | Wildguard    | 0.643  | 0.645     | 0.657       |
> | Grad-DRA     | 0.654  | 0.646     | 0.655       |
> | Repsim-DRA   | 0.642  | 0.645     | 0.658       |
>
> **Table 1:** MMLU performance on retrained Llama models.
>
> | Method      | Male TPR | Female TPR | TPR Gap |
> |-------------|---------:|-----------:|--------:|
> | No Filtering| 0.908    | 0.060      | 0.848   |
> | GradSim-DRA | 0.900    | 0.654      | 0.246   |
> | RepSim-DRA  | 0.860    | 0.768      | 0.092   |
>
> **Table 2:** TPR on retrained Llama models.
>
> We thank the reviewer for raising this. In the jailbreak setting, we additionally evaluate post-filtering utility on MMLU in Table 1, and the results show no observable utility drop after DRA-based filtering while ASR decrease. This indicates that the safety gains are not explained by broad refusal or degraded benign capability. In the gender-bias setting, to make the downstream effect more transparent, we additionally report the underlying group-wise TPRs in Table 2 rather than only the TPR gap. Results show that male TPR is preserved while female TPR increase, leading to less gap between two groups. Thus we believe that the smaller TPR gap is not obtained by uniformly worsening both groups.
>
> > The theoretical contribution is quite weak.
>
> Our main contribution is to formulate representation based unsafe training data detection within a principled theoretical framework, rather than treating similarity scoring as a heuristic choice. We would like to note that **Reviewer fh6D** commented that our framework is a “Clear and Elegant Methodological Design”, while **Reviewer Y8RE** commented that this is a “Conceptually intuitive method.” This formulation lets us identify the key challenge in the task: the unsafe signal can be obscured by noisy or weakly informative feature dimensions. Based on this analysis, we then design DRA to explicitly address these challenges through denoising and dimension selection, without requiring unsafe labels on the training data. The empirical gains in both detection performance and downstream safety show that this theory grounded formulation is not only conceptually meaningful, but also directly useful in practice.
>
> > How is the performance of selecting unsafe training data using powerful LLMs like GPT-5 or Gemini Pro?
>
> | Method          | XSTest | ToxicChat | Gender-bias |
> |-----------------|-------:|----------:|------------:|
> | GPT-5.4           | 0.795  | 0.731     | 0.251       |
> | DRA-Grad  | 0.986  | 0.926     | 0.748       |
>
> **Table 3** Comparison between GPT 5.4 and DRA-Grad on Llama
>
> We additionally evaluate GPT-5.4 under the same setup and report the results in the table above. DRA still performs substantially better: GPT-5.4 achieves AUPRC $\textbf{0.795}$ on XSTest and $\textbf{0.251}$ on Gender-bias, versus $\textbf{0.986}$ and $\textbf{0.748}$ for GradSim-DRA on Llama, respectively. The especially large gap on gender-bias is expected, since this setting is model-aware and GPT-5.4 does not have direct knowledge of how particular training samples affect the downstream model’s behavior, so prompting alone is much less effective for identifying such harmful data.
>
> > Comparison with non-linear dimension reduction methods.
>
> | Method     | XSTest | ToxicChat | Gender-bias |
> |------------|-------:|----------:|------------:|
> | Kernel PCA | 0.986  | 0.929     | 0.662       |
> | Umap       | 0.004  | 0.262     | 0.207       |
> | PCA (ours) | 0.986  | 0.926     | 0.748       |
>
> **Table 4:** Comparison between PCA with non-linear dimension selection methods.
>
> Beyond PCA, we also evaluated nonlinear alternatives such as Kernel PCA and UMAP. Kernel PCA performs similarly to PCA, while UMAP is substantially worse. This supports our view that DRA is limited less by nonlinear expressiveness than by noise in the representation space, including high-variance nuisance directions and weakly separable dimensions. Our method addresses this by whitening to suppress irrelevant variance and then selecting dimensions most informative for unsafe-data detection.

---

> > ### Author Rebuttal · Reviewer_dh4A · 2026-03-31
> >
> > Thank you for the rebuttal. I'll raise the score accordingly.

---

### Official Review · Reviewer_Y8RE · 2026-03-09

**Soundness:** 3
**Presentation:** 3
**Significance:** 3
**Originality:** 3
**Overall Recommendation:** 4
**Confidence:** 4

**Summary:**

This paper studies the problem of detecting unsafe training data in large language models (LLMs). The authors argue that LLMs are sensitive to unsafe training examples, which may lead to harmful behaviors. Existing approaches often rely on moderation classifiers, which introduce additional computational overhead and depend on predefined safety taxonomies that may not generalize well to diverse unsafe behaviors. To address these limitations, the paper proposes Denoised Representation Attribution (DRA) within an unsafe-data filtering framework. The method detects unsafe training examples by measuring the similarity between training samples and a small set of unsafe target examples using internal model representations such as hidden states or gradients. Since these representations may contain noise from irrelevant features, the authors introduce a denoising step before computing attribution scores. Specifically, the method performs a one-time preprocessing step that estimates background statistics from training representations and applies truncated PCA whitening, retaining the most informative components and selecting relevant dimensions. Using these denoised representations, the method computes a detection score that ranks training samples according to their similarity to unsafe target examples. The highest-scoring samples are considered likely unsafe and can be filtered from the training data. The approach is evaluated on two tasks: jailbreaking injection detection and gender bias mitigation. Detection performance is primarily measured using Area Under the Precision–Recall Curve (AUPRC) along with other safety-related metrics. The results suggest that denoising representations before attribution can improve unsafe data detection and contribute to safer retraining outcomes.

**Compliance With Llm Reviewing Policy:**

Affirmed.

**Key Questions For Authors:**

1. How sensitive is the detection performance to the size and diversity of the unsafe target set used for attribution?
2. How does the choice of retained PCA components and selected dimensions affect the detection results?
3. Does the method risk incorrectly flagging safe training samples as unsafe, and how is this controlled?
4. Have the authors evaluated whether DRA generalizes across different model architectures or datasets?

**Limitations:**

The paper would benefit from a clearer discussion of how the method depends on the availability and quality of unsafe target examples, as well as potential limitations when unsafe behavior is subtle or not well represented in the target set.

**Strengths And Weaknesses:**

Strengths:
1. Addresses an important problem: The paper tackles the problem of identifying unsafe training data in LLMs, which is an increasingly important issue as models are trained on large-scale and potentially noisy datasets.
2. Clear motivation: The paper provides a clear motivation for moving beyond moderation classifiers, highlighting their computational overhead and dependence on predefined safety taxonomies.
3. Conceptually intuitive method: The proposed DRA approach is relatively simple and intuitive. By denoising representations before computing attribution scores, the method aims to improve the reliability of representation-based similarity measures.
4. Evaluation across multiple safety scenarios: The method is evaluated on two different safety settings (jailbreaking injection detection and gender bias mitigation), suggesting that the approach may generalize across different unsafe behaviors.

Weaknesses:
1. Limited analysis of the denoising step: The method relies heavily on the denoising process (PCA whitening and dimension selection), but the paper provides little analysis of how sensitive the results are to the number of retained components or selected dimensions.
2. Dependence on unsafe target examples: The framework requires a small set of unsafe target examples. It is unclear how sensitive the method is to the size, quality, or diversity of this target set.
3. Generalizability: The paper explicitly notes that it does not require a fine-grained taxonomy of unsafe behaviors. While this is a reasonable design choice, the paper doesn't provide enough evidence to support the proposed method's ability to distinguish between different types of unsafe content unless it is specifically trained to do so using a target dataset.

---

> ### Author Rebuttal · Authors · 2026-03-30
>
> > Limited analysis of the denoising step. (also Key Questions 2)
>
> | Method      | r = 5 | r = 10 | r = 30 | r = 50 | Early Stop | Random r = 30 |
> |-------------|------:|-------:|-------:|-------:|-----------:|--------------:|
> | DRA-Grad    | 61.8  | 77.5   | 82.3   | 82.3   | 80.7       | 6.5           |
> | DRA-Repsim  | 91.6  | 93.4   | 92.6   | 93.0   | 94.0       | 41.0          |
>
> **Table 1:** Gemma-2-9b ToxicChat experiment results.
>
> We thank the reviewer for this comment. The denoising step in DRA is designed to address exactly this tradeoff: whitening removes high-variance nuisance directions, while the leave-target-out (LTO) criterion selects the most informative remaining dimensions. We used a fixed r=30 across experiments for consistency, and new ablations in Table 1 show that the method is not highly sensitive to this choice: performance improves as r increases, then saturates once the useful dimensions are included. We also test a simple early-stopping rule based on LTO AUPRC and obtain similar results to fixed r=30, while random dimension selection performs much worse. Together, these results suggest that DRA is robust to the precise choice of retained dimensions, and that its gains come from LTO-guided denoising rather than from tuning r.
>
> > Dependence on unsafe target examples. (also Key Questions 1)
>
> We have included sensitivity analysis in the appendix (Figure 3, Appendix C.2), where increasing target-set size and diversity improves detection performance. However, even very small target sets already work well in practice. In particular, as few as 5 target examples are sufficient to achieve strong, state-of-the-art detection results. These results suggest that DRA is not fragile to target-set size.
>
> > Generalizability.
>
> We thank the reviewer for this comment. We note that there is a tradeoff behind our design: unlike classifier-based approaches, DRA does not require a predefined unsafe taxonomy or additional training, and instead operates in a target-conditioned way using only a small target set. In this sense, DRA’s generalizability is not taxonomy-level identification of every unsafe subtype, but the ability to flexibly detect harmful training data for whichever unsafe behavior is specified by the target set. If multiple unsafe behaviors are relevant in practice, one can simply construct multiple target sets and run detection for each, which is often more realistic than assuming access to a complete and fixed unsafe taxonomy.
>
> > Does the method risk incorrectly flagging safe training samples as unsafe, and how is this controlled?
>
> | Method       | XSTest | ToxicChat | Gender-bias |
> |--------------|-------:|----------:|------------:|
> | No Removal   | 0.642  | 0.646     | 0.651       |
> | Wildguard    | 0.643  | 0.645     | 0.657       |
> | Grad-DRA     | 0.654  | 0.646     | 0.655       |
> | Repsim-DRA   | 0.642  | 0.645     | 0.658       |
>
> **Table 2:** MMLU performance on retrained models.
>
> We thank the reviewer for this comment. First, detection quality is measured with AUPRC, which directly captures the precision–recall tradeoff, so a strong score requires not only recovering unsafe samples but also avoiding excessive false positives on safe ones. Second, in retraining we remove only a very small number of top-ranked samples (100 in our setup, less than 1% of the training data), and our added utility evaluation in Table 2 shows that this filtering does not cause utility degradation. Together, these results suggest that DRA is effective at isolating harmful samples while keeping removal of safe data limited.
>
> > Have the authors evaluated whether DRA generalizes across different model architectures or datasets?
>
> We thank the reviewer for this comment. We already evaluate DRA across multiple model architectures and datasets in the paper. Specifically, on both Llama and Gemma models, and on three distinct settings covering two jailbreak datasets (XSTest and ToxicChat) as well as a bias setting (Gender-bias). Across these settings, DRA consistently improves detection quality and downstream safety.

---

> > ### Author Rebuttal · Reviewer_Y8RE · 2026-04-06
> >
> > I appreciate the author's attempt to clarify my concerns.

---

> > > ### Author Response · Authors · 2026-04-07
> > >
> > > We sincerely thank the reviewer for the follow-up and for indicating that their concerns have been addressed. In particular, the rebuttal directly resolves two of the key issues raised in the original review: sensitivity to the choice of retained dimensions $r$, and dependence on the size/diversity of the unsafe target set. Together with the results already in the paper, these additional analyses show that DRA is robust to these design choices, which further strengthens the paper's main contribution. Therefore, we would be very grateful if the reviewer could kindly reconsider the score.

---

### Official Review · Reviewer_fh6D · 2026-03-13

**Soundness:** 3
**Presentation:** 3
**Significance:** 3
**Originality:** 3
**Overall Recommendation:** 4
**Confidence:** 3

**Summary:**

This paper introduces Denoised Representation Attribution (DRA), a representation-based method to find and filter unsafe training data for LLMs. DRA improves similarity-based attribution (like RepSim and GradSim) by denoising representations: it applies PCA, then Mahalanobis whitening, then selects the most “unsafe-sensitive” dimensions using a leave-target-out discriminability score. Similarity is computed only in this subspace; the top-scoring training samples are removed and the model is retrained. Evaluated on jailbreak detection (ToxicChat, XSTest-Response) and gender bias mitigation (Bias in Bios), DRA boosts detection AUPRC and reduces attack success rate and gender-gap metrics after retraining.

**Compliance With Llm Reviewing Policy:**

Affirmed.

**Final Justification:**

I still believe the paper would benefit from a clearer safety-utility framing and a compact algorithm box, but I am satisfied with the response overall. My evaluation remains unchanged.

**Key Questions For Authors:**

See Weaknesses.

**Limitations:**

* The paper does not explicitly state whether D_{target} is assumed to be unimodal or multimodal. However, by aggregating target representations into a single mean vector \mu_Q and using a linear detector, the approach implicitly assumes that the unsafe signal can be captured by a dominant direction in representation space (i.e., primarily through first-order statistics), even if multiple underlying unsafe modes may exist in practice.

* There is no principled strategy provided for choosing K (number of samples to remove).


* The choice of r (selected dimensions) appears fixed (e.g., r = 30), without a clear model selection strategy.

**Strengths And Weaknesses:**

Strengths:
* Clear and Elegant Methodological Design. Methodologically, it is clean and well-motivated: standard tools (PCA, whitening, feature selection) are combined in a principled way, using signal detection theory and justified whitening and selection. The pipeline stays simple, efficient, and effective, turning raw similarity into a Mahalanobis-aware, noise-suppressed score.

* Strong empirical results. DRA delivers clear gains over GradSim and RepSim, especially for jailbreaks, with large ASR drops after retraining. It is tested across multiple models, two safety settings (jailbreaks and gender bias), and both detection (AUPRC) and downstream safety metrics (ASR, TPR gap).

* Practical Impact. The method does not require training new moderation classifiers or additional labeled unsafe data. It is taxonomy-agnostic and can adapt to different unsafe signals defined via the target set. This makes it attractive for data-centric safety workflows.


Weaknesses:

* In Table 3, ASR sometimes increases slightly (e.g., RepSim-DRA on Gemma for ToxicChat), despite strong improvements elsewhere.

* The paper evaluates safety improvements (ASR, TPR gap), but does not evaluate whether filtering harms general model utility. A more informative framing here is the safety-utility Pareto frontier (ASR, Utility, K).

* An algorithm box or pseudocode would improve clarity and reproducibility.

---

> ### Author Rebuttal · Authors · 2026-03-30
>
> > In Table 3, ASR sometimes increases slightly (e.g., RepSim-DRA on Gemma for ToxicChat), despite strong improvements elsewhere.
>
> While DRA does not improve ASR in every single entry of Table 3, this is a small absolute difference of only $\textbf{1.7}$ percentage points, corresponding to a very small number of additional unsafe responses on the evaluation set, and does not change the overall conclusion. Across models and datasets, DRA consistently improves detection quality (Table 1, 5) and reduces ASR (Table 3, 6) in the clear majority of retraining settings, often by substantial margins. We will revise the text in later revisions to make the claim more precise.
>
> > The paper evaluates safety improvements (ASR, TPR gap), but does not evaluate whether filtering harms general model utility. A more informative framing here is the safety-utility Pareto frontier (ASR, Utility, K).
>
> | Method       | XSTest | ToxicChat | Gender-bias |
> |--------------|-------:|----------:|------------:|
> | No Removal   | 0.642  | 0.646     | 0.651       |
> | Wildguard    | 0.643  | 0.645     | 0.657       |
> | Grad-DRA     | 0.654  | 0.646     | 0.655       |
> | Repsim-DRA   | 0.642  | 0.645     | 0.658       |
>
> **Table 1:** MMLU performance on retrained models.
>
> We thank the reviewer for this suggestion. We additionally measure post-filtering utility on MMLU in Table 1. The results show that DRA-based filtering preserves general model utility, while the ASR tables show clear safety gains after retraining. These two results together support the key practical conclusion: DRA removes a small set of highly influential unsafe samples and delivers improved safety without sacrificing overall utility.
>
> > The paper does not explicitly state whether D_{target} is assumed to be unimodal or multimodal.
>
> We thank the reviewer for this thoughtful comment. We agree that the current formulation uses the mean target representation​ and therefore does not explicitly model multiple unsafe modes; we already note this limitation in the paper when discussing more heterogeneous multi-genre unsafe mixtures. At the same time, we would like to clarify that our target sets are not necessarily unimodal in a narrow sense: for example, ToxicChat already contains diverse unsafe behaviors rather than a single category, and our method remains effective in that setting. More importantly, Appendix C.2 shows that increasing target-set variety improves detection performance, especially when the training data contains multiple unsafe genres, which is consistent with the idea that a more diverse target set helps cover heterogeneous unsafe modes. In practice, if multiple unsafe modes are relevant, one can collect a small target set for each mode, or pool them into a more diverse target set, which is realistic in many applications and is compatible with our framework.
>
> > There is no principled strategy provided for choosing K (number of samples to remove).
>
> We thank the reviewer for this comment. Since our method outputs a ranking of training samples, we primarily evaluate detection quality with AUPRC, a threshold-free metric that assesses performance across all choices of K rather than a single removal budget. In practice, K can then be chosen according to the desired filtering budget. Our retraining results further show that a large K is unnecessary: removing only 100 top-ranked samples (less than 1% of the data) already yields clear safety improvements, indicating that DRA remains effective even under conservative filtering budgets.
>
> > The choice of r (selected dimensions) appears fixed (e.g., r = 30), without a clear model selection strategy.
>
> | Method      | r = 5 | r = 10 | r = 30 | r = 50 | Early Stop | Random r = 30 |
> |-------------|------:|-------:|-------:|-------:|-----------:|--------------:|
> | DRA-Grad    | 61.8  | 77.5   | 82.3   | 82.3   | 80.7       | 6.5           |
> | DRA-Repsim  | 91.6  | 93.4   | 92.6   | 93.0   | 94.0       | 41.0          |
>
> **Table 2:** Gemma-2-9b ToxicChat experiment results.
>
> We thank the reviewer for this comment. The role of r in DRA is to balance useful denoised signal against residual noisy dimensions. This is already controlled by the method design: whitening suppresses high-variance nuisance directions, and the leave-target-out criterion selects the most informative remaining dimensions.We used a fixed r=30 across experiments for consistency, and new ablations in Table 2 show this choice is robust: performance increases with r as more useful dimensions are captured and then saturate. In addition, a simple early-stopping rule based on LTO (Leave-Target-Out) AUPRC (stop including more dimension when LTO AUPRC no longer increase, LTO proposed in Section 3.3) gives similar results to fixed r=30, while random dimension selection is much worse. Together, these results suggest that DRA does not depend on a narrowly tuned r, and that the gain comes from the LTO-guided selection rather than from using more dimensions.

---

> > ### Author Rebuttal · Reviewer_fh6D · 2026-04-02
> >
> > Thank you for the detailed rebuttal and the additional results. The clarifications were helpful, especially on utility, K, and the sensitivity to r. I still believe the paper would benefit from a clearer safety-utility framing and a compact algorithm box, but I am satisfied with the response overall. My evaluation remains unchanged.

---

> > > ### Author Response · Authors · 2026-04-07
> > >
> > > We sincerely thank the reviewer for the thoughtful follow-up and for indicating that their concerns have been addressed. In particular, the additional results in the rebuttal directly strengthen two key points raised in the original review: first, DRA is robust to important design choices rather than relying on narrow tuning. Second, the added MMLU results show that the safety gains do not come at the cost of general utility. Together, these clarifications strengthen the practical applicability of the method and the completeness of the evaluation. Therefore, we would be very grateful if the reviewer could kindly reconsider and raise the score.

---

### Decision · Program_Chairs · 2026-04-30

**Decision:**

Accept (regular)

**Comment:**

This paper identifies a fundamental limitation in existing unsafe data detection methods for large language models, namely the noise introduced by neutral tokens within unsafe examples that degrades attribution quality. To address this, it proposes Denoised Representation Attribution (DRA), a novel framework that refines representation-based data attribution by filtering out irrelevant components in both training and target samples. The approach avoids reliance on predefined taxonomies and expensive classifier training, offering a more flexible and efficient alternative to moderation-based methods. Empirical results on tasks such as jailbreak filtering and bias detection demonstrate that DRA consistently outperforms state-of-the-art approaches, supporting its practical effectiveness for safer LLM development. During the rebuttal, the authors successfully addressed the reviewers' concerns, and all reviewers agreed to accept this work. We strongly recommend that the authors incorporate the reviewers’ feedback and the additional results presented in the rebuttal into the final version.

Congratulations!